# Unlicensed GS-441524-Like Antiviral Therapy Can Be Effective for at-Home Treatment of Feline Infectious Peritonitis

**DOI:** 10.3390/ani11082257

**Published:** 2021-07-30

**Authors:** Sarah Jones, Wendy Novicoff, Julie Nadeau, Samantha Evans

**Affiliations:** 1Department of Veterinary Biosciences, College of Veterinary Medicine, The Ohio State University, Columbus, OH 43210, USA; Sarahejones79@yahoo.com; 2Departments of Orthopaedic Surgery and Public Health Sciences, School of Medicine, University of Virginia, Charlottesville, VA 22903, USA; wmn2v@virginia.edu; 3Hamilton Region Veterinary Emergency Clinic, Hamilton, ON L8P 4W3, Canada; julie.nadeau7@gmail.com

**Keywords:** FIP, GS-441524, COVID-19, black market, coronavirus, cat, remdesivir

## Abstract

**Simple Summary:**

Feline infectious peritonitis (FIP) is a fatal disease of cats caused by feline coronavirus. The goal of this study was to formally evaluate the administration of unlicensed, crowd-sourced antiviral GS-441524-like therapy for cats suspected to have feline infectious peritonitis (FIP). Members of a large social media support and GS-441524-like drug distribution group were surveyed via the Internet. Of the 393 analyzed surveys which met inclusion criteria, 73.7% of owners utilizing this therapy were from the United States. Only 8.7% of owners reported receiving significant help from their veterinarian in treating their cat. The mean cost of treatment was USD 4920. A majority of owners (88.2%) reported noticeable improvement in clinical signs within one week of initiating therapy. At the time of the survey, 96.7% (380 cats) were alive, with 54.0% of them considered cured and another 43.3% being monitored in the 12-week observation period. A total of 12.7% of the cats suffered a relapse of clinical signs of FIP, and 3.3% of the cats died despite GS-441524-like therapy. Reported complications were mostly related to owner administration of subcutaneous injections of the acidic GS-441524-like therapy, such as vocalization, pain, struggling, and injection-site wounds. Limitations of this study include a retrospective design, bias in case selection, reliance on owner-reported data, and inability to confirm the contents of unlicensed pharmaceuticals; however, important lessons can be learned from the experiences of these owners. While unconventional, and certainly not free from medical and legal risks, unlicensed, at-home GS-441524-like therapy, according to owner reports, can apparently offer benefits in the treatment of cats suspected of FIP.

**Abstract:**

The goal of this study was to formally evaluate the administration of unlicensed, crowd-sourced antiviral GS-441524-like therapy for cats suspected to have feline infectious peritonitis (FIP), a previously fatal disease. Members of a large social media support and GS-441524-like drug distribution group were surveyed via the Internet. The survey was targeted toward owners who had treated their cats for at least 12 weeks with unlicensed GS-441524-like drugs. Of the 393 analyzed surveys which met inclusion criteria, 73.7% of owners utilizing this therapy were from the United States. Only 8.7% of owners reported receiving help from their veterinarian in administering the treatment to their cat. The mean cost of treatment was USD 4920. A majority of owners (88.2%) reported noticeable improvement in clinical signs within one week of initiating therapy. At the time of the survey, 96.7% (380 cats) were alive, with 54.0% of them considered cured and another 43.3% being monitored in the 12-week observation period. A total of 12.7% of the cats suffered a relapse of clinical signs of FIP, and 3.3% of the cats died despite GS-441524-like therapy. Reported complications were mostly related to owner administration of subcutaneous injections of the acidic GS-441525-like therapy, such as vocalization, pain, struggling, and injection-site wounds. Limitations of this study include a retrospective design, bias in case selection, reliance on owner-reported data, and inability to confirm the contents of unlicensed pharmaceuticals; however, important lessons can be learned from the experiences of these owners. While unconventional, and certainly not free from medical and legal risks, unlicensed, at-home GS-441524-like therapy, according to owner reports, can apparently offer benefits in the treatment of cats suspected of FIP.

## 1. Introduction

Since March of 2020, the spread of the coronavirus known as SARS-CoV2 has developed into the global COVID-19 pandemic. A related, but feline-specific, virus known as feline coronavirus (FCoV) is the causative agent of a fatal disease of cats known as feline infectious peritonitis (FIP). Antiviral drugs that inhibit viral RNA replication have offered potential therapeutic and preventative benefits for COVID-19, including the adenosine nucleoside monophosphate prodrug GS-5734, more commonly known as remdesivir (marketed by Gilead Sciences, Inc. (Foster City, CA, USA)) [1]. A 2018 study published by Murphy et al. demonstrated that GS-441524, a less chemically complex parent nucleoside (also patented by Gilead), was highly effective against experimentally induced FIP at a dosage of 4.0 mg/kg subcutaneously every 24 h for 12 weeks (84 days) in 10 laboratory cats [2]. Later in 2019, the same group conducted a clinical trial using GS-441524 to treat 31 client-owned cats with FIP, and the results indicated that this once fatal disease was clinically reversible with this nucleoside analog [3]. The compound name for GS-441524 is (2R,3R,4S,5R)-2-(4-Aminopyrrolo[2,1-f][1,2,4]triazin-7-yl)-3,4-dihydroxy-5-(hydroxymethyl)tetrahydrofuran-2-carbonitrile, which has been known by a variety of other synonyms [4]. For brevity in this report, we refer to multiple unlicensed drug formulations which claim to contain this compound as ‘GS-441524-like’, even though they were not produced by Gilead Sciences.

FCoV (feline coronavirus) is an extremely common virus, to the point of being considered ubiquitous worldwide [5], with seroprevalence of between 26 and 87% (depending on the study and the region of the world) in cats originating from multi-cat environments (catteries or breeders and animal shelters) [6]. Active infections with FCoV cause mild to subclinical transient GI infections in most patients [7,8]. While 70% of these cats are transiently infected with FCoV, 13% remain persistently infected and chronically shed the virus in their feces [9]. FIP arises from a mutated form of FCoV in approximately 5% of these cats [8] and kills an estimated 0.3 to 1.4% of cats worldwide annually [3]. The two major clinical presentations of FIP include an effusive (“wet”) form, in which the patient exhibits cavity effusions, and a non-effusive (“dry”) form. Until very recently, the development of FIP in cats was uniformly fatal and often challenging to accurately diagnose antemortem [10].

Possibly due to the desire to continue to explore human RNA viruses that might be inhibited by GS-441524, Gilead halted further drug development of GS-441524 therapy in cats. This created a demand for GS-441524 from cat owners whose cats were diagnosed with FIP and facing a certain death. As a result, multiple unlicensed (sometimes referred to as “black market”) drug manufacturers began selling this compound to cat owners over the Internet, which is not currently authorized under the United States patent by Gilead Sciences [4]. The evolution continued into a large-scale, crowd-sourced, online FIP treatment and support network, the likes of which have never before been encountered in veterinary medicine. There are now several large social network groups that help cat owners obtain GS-441524-like drugs, plan a protocol, and treat their cats with suspected FIP. As GS-441524 remains an unlicensed therapy in the United States and elsewhere, the majority of this drug is produced, purchased, administered, and monitored without much, if any, veterinary oversight.

The goal of this study was to formally evaluate the administration of this unlicensed, crowd-sourced therapy for FIP and determine what factors may lead to treatment success. To do this, we surveyed members of the largest of the aforementioned social media support and GS-441524-like drug distribution groups. What we discovered was truly astounding; not only are lay people routinely attempting this therapy on their cats, but it is consistently effective in achieving a cure according to owner-reported data. Additional patient, treatment, and outcome characteristics are analyzed in this report.

## 2. Materials and Methods

A survey (included as Appendix A) was created through Google Forms (docs.google.com/forms). The survey was circulated widely on a prominent social media page (“FIP Warriors”) dedicated to GS-441524-like treatment and distribution. It was advertised and targeted toward owners whose cats were suspected to have FIP (based on variable criteria used by their individual veterinarians) and had received at least some amount of treatment with GS-441524. Some of these patients had received the full 84 days of treatment, while others had begun the treatment protocol and subsequently died of the disease or were euthanized due to complications. Inclusion criteria were survey submissions regarding any cat that had undergone GS-441524-like treatment for suspected FIP and had completed at least 84 days of treatment (regardless of the outcome thereafter) or that had begun treatment and subsequently died. Exclusion criteria were survey submissions regarding patients that had undergone <84 days of therapy (due to being mid-treatment protocol) or that were missing a majority of data from unanswered survey questions.

The survey took cat owners approximately thirty minutes to complete, and questions consisted of both multiple choice and short answer format. Some questions appeared based on previous answers, in a hierarchical arrangement, as owners worked their way through the survey. Permission to share and circulate the survey was obtained from the social media group’s administrators, and it was frequently shared and posted on all affiliated pages. Survey data were collected between 31 July 2020 and 25 October 2020. The study was approved by the Ohio State University’s Institutional Review Board (protocol #2021E0162).

All data were entered into Minitab version 19 (State College, PA, USA) for analysis. Descriptives and frequencies for all categorical variables were computed. Basic descriptive statistics were calculated for all continuous variables. Two-sample *t*-tests or one-way ANOVAs were used to compare differences between groups, with alpha set at 0.05 and 80% power.

## 3. Results

### 3.1. Owner/Patient Characteristics

A total of 411 surveys were collected during the study period, with 393 cases meeting inclusion criteria and having a known status (alive vs. deceased). The remaining 18 surveys were not complete enough to obtain a sufficient amount of information. Of those 393 cases, 64.9% were male (52.3% neutered males and 12.6% intact males) and 35.1% were female (25.9% spayed females and 9.2% intact females).

The majority of owners were from the United States (73.7%), with the next highest number coming from Canada (6.6%) (Table 1). The mean age at diagnosis was 1.65 years (approximately 1 year and 8 months) with a standard deviation of 2.28 years and a range of 2 months to 18 years. More than two thirds of the cats (67.2%) were reported to be multiple breeds, domestic longhairs, or domestic shorthairs. The remaining 32.8% were a wide variety of pure breeds, with the most common breed reported as Siamese (5.6% of all cats) (Table 2). This supports a 2006 sixteen-year retrospective study by Pesteanu-Somogyi et al. that found purebred cats to be significantly more likely to be diagnosed with FIP [11]. Almost one fourth of owners reported that their veterinarian either directly (14.5%) or indirectly (12.0%) told them about unlicensed GS-441524, while 30.36% of owners said they found the information from doing their own online research, and 23.21% said they found out about GS-441524-like therapy on social media webpages for owners of cats with FIP (Table 3).

While a majority of owners (291 responses, 74.3%) reported having at least some help from a veterinarian (usually involving help with diagnostic monitoring), only 34 (8.7%) reported receiving significant help in administering therapy from their veterinarians, such as assistance with drug injections or oral compound administration. Of note, 101 (25.7%) owners reported receiving no veterinary help other than initial diagnostics. These owners treated their cat entirely on their own, which raises the concern for animal welfare if owners are untrained or inexperienced in injecting their cats or managing post-injection wounds. The mean reported cost across the various brands for treatment with GS-441524-like therapy was USD 4920, with a standard deviation of USD 3115, and a median of USD 4000. The reported cost ranged from a low of USD 500 to a high of USD 21,000.

### 3.2. Disease Characteristics

A majority of owners (224 responses, 57.0%) reported that their cat had signs of effusive (“wet”) FIP, often with grossly noticeable effusions observed by owners. A large contingent (169 responses, 43.2%) of owners reported signs of neurological and/or ocular disease. The most common clinical signs were lethargy/listlessness (347 responses, 88.3%), decreased appetite (308 responses, 78.4%), and fever (258 responses, 65.7%) (Table 4).

The most common signs of response to therapy were improved appetite (287 responses, 73.8%), improved movement/walking/jumping (205 responses, 52.7%), improved energy (153 responses, 39.3%), and resolution of fever (104 responses, 26.7%).

Among the 50 cats that had a reported relapse (12.7%), more than half (69.0%) developed worsening signs related to appetite, lethargy, weakness, and fever. An additional 14.3% of owners observed neurological or ocular signs.

### 3.3. Treatment Characteristics

There were many different unlicensed brands of GS-441524-like therapy reported. The country of origin of these compounds is not readily available. Mutian was reported most often (19.4%), followed by owners that used multiple different brands during treatment (18.1%), largely because the original brand was no longer available when they needed to re-order (Table 5). Possibly due to the common confusion among survey respondents about the difference in “dosage” (mg/kg) versus “dose” (total mg), only 203 owner responses accurately reported starting and ending dosages for their cats. The mean starting dosage was 6.7 mg/kg (range 2 mg/kg to 16 mg/kg) with a standard deviation of 2.24 and a median of 5 mg/kg, and the mean ending dosage was 8.5 mg/kg (range 3 mg/kg to 25 mg/kg) with a standard deviation of 3.34 and a median of 8 mg/kg. Starting dosage was loosely based on a previous study out of UC Davis [3], and owners sought help from the website administrators of the social media network for individual changes based on patient response and recent laboratory findings. Of note, no dosages were reportedly decreased over the treatment period. Using a paired *t*-test, the difference between starting and ending dosage was highly significant (*p* < 0.001). This reported dosage differs from the 2018 study that found 4.0 mg/kg subcutaneously every 24 h for a minimum of 12 weeks to be the optimum dose [3]. Starting and ending dosages were significantly different for cats with owner-reported neurological or ocular signs compared to cats that did not have neurological or ocular involvement (mean dosage of 8.01 mg/kg versus 5.44 mg/kg at the start of treatment; mean dosage of 10.47 mg/kg versus 6.62 mg/kg at the end of treatment).

Out of the 393 patients, 348 (88.5%) were administered 12 weeks (84 days) of treatment, with 45 patients (11.4%) receiving extended treatment (with an average of 4 extra weeks of therapy). Treatment extension was based on whether signs had resolved completely at the end of the standard 12 weeks of therapy and does not include cats that relapsed during or after the observation phase. The majority of patients (71.7%) were treated with injectable GS-441524, with 8.2% receiving oral formulations, and 20.1% receiving a combination of oral and injectable formulations. Treatment with other medications and supportive therapies was also recorded as part of the survey (Table 6). Among the most common supplemental therapies were vitamin B1 or B12 (41.8%) and the use of steroids (37.9%). Only 8.2% of owners reported using gabapentin.

The owners of a majority of patients receiving injections reported vocalization at the time of injection (309 responses, 82.0%) and/or pain at the injection site (287 responses, 76.1%). More than half (203 responses, 51.7%) reported scarring and scabbing at the injections site(s) (Table 7).

### 3.4. Outcomes

For the cats that met inclusion criteria at the time of the survey, 380 (96.7%) were alive and 13 (3.3%) had died. With regard to the current phase of treatment, 54.0% were reported as “cured,” 43.3% were in the observation period (having survived at least 12 weeks of treatment), and 2.7% were being treated again for a relapse of FIP signs. Among the 224 cats with signs of effusive (“wet”) FIP, eight cats (3.6% of cats in this group) died by the time the owner filled out the survey. Among 169 cats with non-effusive (dry) FIP, five (2.96%) died by the time the owner took the survey. Among the 169 cats reported to have neurological or ocular signs, eight (4.7% of cats in this group) died. Four of those cats that died had both neurological/ocular and effusive (wet) FIP signs, and four cats that died had both neurological/ocular and non-effusive (dry) FIP signs. The difference in mortality between these groups was not statistically significant.

Most owners (88.2%) reported noticeable improvement within one week of initiating GS-441524-like treatment, and the mean time to “return to normal behavior” was 3.85 weeks (range 1 to 16 weeks). Owners of cats with either neurological or ocular signs reported that 66% of them responded within one week, and 16.3% of those cats showed marked improvement within one day. A relapse of suspected FIP (defined as recurrence of signs following the end of the 12-week treatment period) was reported in 50/393 cases (12.7%), with 48 of those 50 cat owners (96%) reporting only a single relapse. Five cats (10%) with reported relapse died or were euthanized, all after a single relapse. Among the 45 cats (90%) with a relapse of FIP that were alive at the time of the survey, nine were currently in treatment, 17 were in the 12-week observation period, and 19 were considered cured. Four (8%) of the fifty cats with reported relapse had reported neurological or ocular signs at the time of diagnosis; this was not a statistically significant difference from the number of cats without neurological or ocular signs at the time of diagnosis.

## 4. Discussion

Here, we present the first report in the scientific literature of unlicensed antiviral therapy used to successfully treat feline infectious peritonitis (FIP) by cat owners at home. To the authors’ knowledge, this is the first widespread use of any unlicensed, crowd-sourced drug to treat a veterinary disease. We are not advocating for unauthorized use of this medication; our goal is to present this information to better inform cat owners and veterinarians who are facing decisions about the treatment of suspected FIP.

Male cats were overrepresented in our study (64.9%), similar to previous findings by Riemer et al. that showed male sex was significantly correlated with FIP [12]. Close to three fourths (72.7%) of the owners answering the survey were based in the United States. This could be due in part to the survey being written in English, and while the social media site targeted here is an international platform, the members appear to be based mostly in the United States. The mean age of the cats in our study was one year and eight months, which corresponds to Riemer et al. (2016), who found that FIP occurred significantly more often in cats under two years of age [12]. More than two thirds (67.2%) of the cats reported were multiple breed or domestic cats, which is slightly lower than what Riemer et al. reported for cats with FIP in 2016 (78.8%) [12]. The survey identified that 32.8% of the cats receiving treatment with GS-441524-like drugs were purebred, with Siamese (5.6%), Maine Coon (3.05%), Bengal (2.8%), and British Shorthair (2.54%) being the most commonly reported breeds. The overrepresentation of purebred cats reported in our study could be due to an increased prevalence of FIP, as previously reported by Pesteanu-Somogyi et al. (2006) [11]; due to being housed in multi-cat environments at a higher rate; or possibly due to owners of purebred cats being, on average, more affluent or financially invested in their cats.

Interestingly, approximately one fourth (26%) of owners reported treating their cats entirely on their own following a diagnosis of FIP, without the help or knowledge of their veterinarians. Of note, although the majority of our survey respondents did not receive help from their veterinarians with therapy, we suspect that the vast majority received a presumptive diagnosis of FIP via a veterinarian, as clinical and laboratory data (for example, complete blood count, biochemistry, fluid analysis, etc.) are required by the social media website administrators before starting therapy. These owners were able to find the support needed to administer GS-441524-like therapy at home from online help through social media websites, particularly from the group’s administrators. Only 8.7% reported that their veterinarian was of significant help to them in treating their cat with GS-441524, by helping to administer the compound instead of simply offering routine exams and repeat diagnostic work-up. A majority of owners (65.6%) still followed up with their veterinarian for care, which usually consisted of repeating blood work and other diagnostics. It is unclear from our results what proportion of those following-up with their veterinarians told the veterinary team about at-home treatment with GS-441524. Anecdotally, social media website administrators are known to track each individual cat’s progress through the treatment protocol on a spreadsheet. They monitor various parameters such as weight, hematocrit, white blood cell count, lymphocyte count, neutrophil count, monocyte count, albumin, globulin, serum A:G ratio (albumin globulin ratio), total bilirubin, BUN (blood urea nitrogen), creatinine, and ALT (alanine transaminase). Based on the progress of monthly blood work values, owners can consult with their assigned website administrator whenever the attending veterinarian is unable to help, either due to limited knowledge of the treatment or an unwillingness to help out of fear of risk to their medical license. This finding brings up important questions about whether the administrators, in their efforts to help FIP patients, could be charged with practicing veterinary medicine without a license and may be unknowingly causing harm, in addition to what role veterinarians can and should have in this endeavor.

Initial anecdotal reports of owners acquiring this medication from unlicensed sources in desperation to help their sick cats came with rumors of a USD 10,000–20,000 price tag, and some of our survey respondents reported paying these prices. However, while certainly still an expensive treatment, owners in our survey reported a mean cost of just under USD 5000 (USD 4920). This difference could be due in part to an increase in unlicensed companies producing the product, driving down the costs through competition. We suspect that the cost of treatment has decreased over time for this reason but cannot definitively determine this, as our survey did not ask about the calendar date of therapy.

Mutian, an international-sourced brand and probably the most well-known GS-441524-like formulation available, does not list GS-441524 on its label and is instead touted as a “dietary supplement” for treating FIP [13]. However, this was the brand of GS-441524-like compound most often reported by owners, suggesting that both owners and website administrators believe this brand contains a GS-441524-like drug. Owners taking the survey reported higher starting dosages than that reported in the original study (4.0 mg/kg) [3] and even higher ending dosages, presumably based on their cat’s response to therapy. The average starting dosage was 6.7 mg/kg, and the average ending dosage was 8.5 mg/kg; this difference was statistically significant (*p* < 0.001). Owners with cats that presented with neurological or ocular FIP signs were encouraged by administrators of the social media website to start at higher dosages (8 mg/kg). It is unclear whether this practice is justified, based on a lack of published evidence surrounding use of GS-441524-like therapy in neurological or ocular FIP, apart from a single case report of ocular disease which was treated with 8 mg/kg of Mutian and simultaneous feline interferon omega [14]. Cats who were initially diagnosed with non-neurological/non-ocular FIP and who later developed neurological or ocular signs were also advised to increase their dosage. Many owners also increased the dosage if their cat was not significantly improving at the 4 or 8-week time point, anecdotally defined as a serum A:G ratio of 0.5 or below. From our observation, some owners choose to start on the lower end of the dosage range initially to see if that will be sufficient to treat their cats’ clinical signs, particularly if cost is of concern. The original studies on GS-441524 for FIP reported lower success in resolving FIP with cats starting at 2.0 mg/kg and determined the optimum dosage to be 4.0 mg/kg, given subcutaneously every 24 h [3]. The difference in effective dosage between the controlled study from UC Davis [3] and those reported by owners in our study who obtained unlicensed GS-441524-like therapy could be due to lack of quality control from the various underground manufacturers. It is possible that unlicensed formulations contain a less pure compound or are otherwise less potent than the pharmaceutical-grade compound. Thus, it is challenging to compare these dosages directly.

Based on the original treatment protocol used in the UC Davis study, the recommended duration of treatment is 84 days (12 weeks) [3]. From our observations on the social media website, the decision to end treatment and begin the 84-day (12-week) observation period is based not only on complete resolution of clinical signs reported by the owner (with rare exceptions for minor residual ocular or neurologic signs) but also on week 12 blood work demonstrating normalization of serum protein levels. Often, a volunteer veterinarian on the social media page will review the blood work for these owners and approve cessation of GS-441524-like therapy. While some cats appear to achieve clinical remission much sooner than the 12-week mark, owners and website administrators are reluctant to discontinue therapy earlier than the 12 weeks out of fear that the cat will relapse. Treatment courses shorter than 12 weeks have not yet been formally evaluated. No survey respondents in our study reported early (<12 weeks) termination of therapy, apart from when the cat died or was euthanized.

The protocol established by the social media website administrators advises all owners to begin treatment with the injectable form of GS-441524-like drug. This is thought to be superior due to concerns over enteric absorption of drugs in cats with gastrointestinal signs (vomiting, diarrhea) or anorexia. After two to four weeks of injectable treatments, the cats are reassessed on a case-by-case basis to see if they can be switched to an oral formulation. There are patient compliance issues with both forms of the medication. The concentration of the compound in the oral formulation is low, requiring multiple capsules to be given at a time, which can be challenging for owners to accomplish. While it is easier to ensure the patient receives the full dose with the injectable form, the pH required to keep the medication stable is very acidic at a pH of 1.5 [3], which likely leads to pain and vocalization upon injection. The majority of owners in our survey (71.7%) reported using only the injectable form of GS-441524, while only 8.2% reported using only an oral formulation. Presumably due to the acidic nature of the injectable form, a majority of patients (89% out of those receiving injections) vocalized following injection, and most (83% out of those receiving injections) also appeared painful at the injection site(s). Pain and morbidity associated with the injection site accounted for a majority of the reported adverse events from the medication. Many cats (46% out of those receiving injections) developed sores or wounds at previous injection sites, and 43% of cats (out of those receiving injections) were reported to bleed at the injection site.

Injectable and oral forms of vitamin B1 or vitamin B12 were the most frequently reported supportive therapy utilized by owners. This was likely intended to help cats that were anemic, had gastrointestinal signs, and/or had anorexia [15]. In addition to the hematological effects, vitamin B12 affects other body systems such as neurological and cardiovascular systems and may aid in preventing disease in humans [16], which in cats with suspected FIP could be used either therapeutically or prophylactically. Not surprisingly, the second most common adjunctive therapy was steroid (presumed glucocorticoid; not specified in the survey) administration, which has traditionally been a part of the standard of care for FIP [17]. A variety of nutraceuticals were reported, and either subcutaneous or intravenous fluid therapy (27.4%, not distinguished by our survey) was commonly utilized as supportive care. The frequent use of antibiotics (presumed due to injection site wounds/abscesses, prophylactic use due to neutropenia, or for other empiric use) raises important concerns about how these drugs were obtained and whether appropriate antimicrobial stewardship was applied. Interestingly, “light therapy” (also known as photobiomodulation) was used by 11.9% of the owners. This therapy is occasionally used in veterinary medicine as an alternative modality to treat inflammatory conditions, provide analgesia, and to accelerate wound healing [18,19]. Light therapy is suspected to have been used mostly to treat the wounds that developed as a result of the injectable form of the treatment. Pain control was not reported as frequently as anticipated. NSAIDs (non-steroidal anti-inflammatory drugs) were rarely mentioned, possibly due to frequent concurrent steroid use. Gabapentin was reported by only 8.2% of owners, and this drug might have been used for either patient sedation/chemical restraint or analgesia.

What we found most striking from these data was the number of cats that reportedly responded well to treatment and, at the time of survey submission, were reported by their owners to be in complete clinical remission (free of clinical signs previously associated with a diagnosis of FIP). Only a small percentage were continuing treatment due to a relapse (2.7%), which is defined as a recurrence of clinical signs that lead to a diagnosis of FIP or emergence of new clinical signs suspicious of FIP. This was true for a variety of clinical manifestations of FIP, including effusive, non-effusive, ocular, and neurological FIP. Nearly all (96.7%) cats were alive when their owners submitted the survey. Previously, it was believed that while GS-441524 was proving to be an efficacious treatment for effusive and non-effusive forms of FIP, ocular and neurological forms were much harder to effectively cure with this treatment due to its inability to cross the blood/eye and blood/brain barriers [2]. Furthermore, previous measurements of GS-441524 in aqueous humor and cerebral spinal fluid (CSF) samples showed substantially lower concentrations than serum samples following subcutaneous injections of GS-441524 [2]. In our study, 42.3% of owners reported that their cat had either neurological or ocular signs, and only 4.7% of these cats reportedly did not survive. Therefore, while more research is needed in this area, our study provides evidence that GS-441524-like therapy is an option for those patients as well. This may be because both the starting and ending dosages of cats in this group treated at home by their owners was substantially higher than the originally published dosages [3].

Due to a plethora of different unlicensed brands of GS-441524-like antiviral drugs, with many owners that used multiple brands throughout the course of treatment, our study was not empowered to discover whether one brand was more efficacious than another. This was compounded by the fact that the majority of cats in the study were alive at the time of writing, suggesting very high efficacy across most, if not all, unlicensed GS-441524-like brands. While we acknowledge the inherent biases in this study, despite the lack of oversight of GS-441524-like drug production, it nonetheless appears to be highly successful in treating FIP according to owner-reported data. Still, it is reasonable to believe that these various formulations may differ in their concentration, purity, pH, potency, absorption pattern, adverse effects, and other characteristics, making this a valuable avenue for further study. Of note, we cannot directly confirm that any of the brands listed here contain any GS-441524-like compound; this is an important area of active investigation for our group and others. It remains possible that some of these brands contain other nucleoside analog antivirals, the protease inhibitor GC376 [20], or something else entirely.

One obvious limitation of our study is that there is no definitive antemortem diagnostic test for feline infectious peritonitis (outside of invasive surgery and biopsy, which is very rarely performed), and our survey respondents self-selected into both joining the FIP social media networks and responding to our survey. Therefore, while 380 of the 393 cats whose owners took the survey were reported to be alive, there is no way to confirm the diagnosis of FIP in these cats or in the 13 cats that died, as no necropsies were reported. Undoubtedly, some of these patients may have been misdiagnosed and their owners misdirected in treating them with GS-441524-like therapy. Thus, a positive response to therapy could have been a mere coincidence in some patients. Similarly, those cases that were reported not to respond well to GS-441524-like therapy (i.e., the 13 cats that died in this study) may have been because the patient was actually suffering from another disease. However, given the cost of this treatment and the great lengths required to obtain it, it is reasonable to assume that, for many patients, most other avenues of diagnostics and therapy had already been pursued and exhausted, and that FIP was indeed the most likely differential. Of note, while there is currently no single, minimally invasive, definitive diagnostic test for FIP, response to treatment with GS-441524-like therapy could be considered supportive or even diagnostic for FIP. A majority of owners (88.2%) reported a dramatic improvement in clinical signs within one week of beginning treatment, with an average time for “return to normal behavior” of 3.85 weeks. Therefore, in the future, cats suspected of having FIP could begin GS-441524-like treatment, and rapid response to the medication could be useful both diagnostically as well as therapeutically. However, it is important to ensure that renal and hepatic function are continually assessed, given that nucleoside analogues may be both nephrotoxic and hepatotoxic [21].

Related to the limitation above, selection bias likely played a large role in the apparent success of therapy in our study. The survey was circulated on the main social media FIP support page, as well as subcategory pages for cats that were considered either “cured”, in relapse and extended treatment, suffering from neurological and ocular FIP, or suffering from effusive (wet) FIP. Survey responses were solicited from owners whose cats had completed at least 12 weeks of treatment. While this accomplished our goal of acquiring information from Internet users of unlicensed GS-441524-like drugs, the results undoubtedly have some amount of bias against owners whose cats began treatment with GS-441524-like therapy but subsequently died of FIP, as they may have left the social media pages or elected to not fill out the survey due to negative emotional states, including lack of enthusiasm for the failed treatment. It is reasonable to assume that these owners are more likely to remove themselves from the webpage after losing their cat to the disease, or that the social media website algorithm may not display these posts to them as often. Further exacerbating this effect is the fact that the original FIP treatment social media webpage was removed by the parent social media company in August of 2020. The remaining group members are likely affected by ‘enthusiasm bias’, whereby only the most enthusiastic and positive-leaning group members remain. These factors are very likely to inflate the success rates reported here; however, a recent study in China found similar, very high survival in GS-441524- and GC376-treated cats (29/30 cats clinically cured) [22]. Thus, these owner-reported outcome data should be considered preliminary, and further investigation into the true efficacy of GS-441524-like therapy under field conditions is urgently needed.

Other limitations of the study include that the survey took around 30 min to complete, at which point owners may have been fatigued from answering the questions. Primacy bias may also have played a strong role in questions that provided long lists of possible answers [23]. Furthermore, relying on owners (with unknown medical backgrounds) to remember the events of a stressful time when caring for a sick cat may lead to inaccurate recall bias. In part because of these issues, our group is pursuing ongoing prospective analysis of unlicensed GS-441524-like therapy for FIP. Still, the data provided here demonstrate an unprecedented event in veterinary medicine, and meaningful lessons can be learned from the experiences of these owners.

Importantly, although GS-441524 is structurally very similar to remdesivir, and there is good reason to believe that it may be effective against other coronaviruses, no survey respondents mentioned use of unlicensed GS-441524-like formulations intended for cats by themselves or others for treatment of humans with SARS-CoV-2. This was slightly surprising given that the same desperation for a treatment for FIP in cats that led to the development of the underground GS-441524-like therapy market could conceivably be found in millions of families across the world with a loved one suffering from COVID-19 during the survey period. However, we have no knowledge of such use, either from our survey data or from anecdotal monitoring of these social media websites. From a regulatory standpoint, unlicensed use of GS-441524-like drugs in humans would likely involve a higher level of legal concern.

## 5. Conclusions

While unconventional, and certainly not free of medical and legal risks, unlicensed, at-home GS-441524-like therapy is reported by owners to effectively abrogate clinical signs in cats suspected of FIP. This unlicensed treatment is apparently efficacious for various clinical presentations of the disease (effusive, non-effusive, neurological, and ocular) according to owner-reported data. The future of this particular drug for the treatment of FIP is unclear, as the status of GS-441524 production, distribution, and patent rights continues to evolve. However, this treatment provides a test-case of social media and crowd-sourced entities rising to fill a void left by the corporate dismissal of a ‘miracle cure’ for a previously 100% fatal disease. This may be the first of such an occurrence in veterinary medicine, but in today’s highly connected Internet age, it almost certainly will not be the last.

## Figures and Tables

**Table 1 animals-11-02257-t001:** Country of origin of survey respondents.

Country	Count of Responses	Percent
United States	289	73.72
Canada	26	6.63
United Kingdom	15	3.83
Italy	13	3.32
Bulgaria	8	2.04
Crostia	4	1.02
Netherlands	4	1.02
Romania	4	1.02
Australia	3	0.77
Ireland	3	0.77
China	2	0.51
Germany	2	0.51
Hong Kong	2	0.51
Mexico	2	0.51
New Zealand	2	0.51
Poland	2	0.51
Portugal	2	0.51
Czech Republic	1	0.26
Hungary	1	0.26
Malaysia	1	0.26
Peru	1	0.26
Serbia	1	0.26
Singapore	1	0.26
Sweden	1	0.26
Switzerland	1	0.26
Turkey	1	0.26
Total	392	100%
Not Reported	1	

**Table 2 animals-11-02257-t002:** Breed reported for cats treated with GS-441524.

Breed	Count of Responses	Percent
Multiple Breeds	264	67.18
Siamese	22	5.6
Maine Coon	12	3.05
Bengal	11	2.8
British Shorthair	10	2.54
Siberian	9	2.29
Ragdoll	8	2.04
Devon Rex	7	1.78
Persian	7	1.78
Russian Blue	7	1.78
Scottish Fold	5	1.27
Sphynx	5	1.27
Birman	4	1.02
Exotic Shorthair	3	0.76
Himalayan	2	0.51
Norwegian Forest Cat	2	0.51
Snowshoe	2	0.51
Abyssinian	1	0.25
Balinese	1	0.25
Bombay	1	0.25
Burmese	1	0.25
Lykoi	1	0.25
Mandalay (Bombay)	1	0.25
Munchkin	1	0.25
Ocicat	1	0.25
Oriental Shorthair	1	0.25
Ragamuffin	1	0.25
Selkirk Rex	1	0.25
Singapura	1	0.25
Skookum	1	0.25
Total	393	100.0

**Table 3 animals-11-02257-t003:** How cat owners first learned about the existence of GS-441524 therapy for FIP. Owners were able to list more than one source, so the total number of responses is higher than the total number of respondents.

Method	Count of Responses	Percent
Website, not Facebook	119	30.36
Facebook	91	23.21
Veterinarian (directly)	57	14.54
Friend	56	14.29
Veterinarian (indirectly)	47	11.99
Rescue	12	3.06
Breeder	8	2.04
Conference	2	0.51

**Table 4 animals-11-02257-t004:** Clinical signs reported in cats treated with GS-441524. Owners were able to list more than one source, so the total number of responses is higher than the total number of respondents.

Clinical Sign	Count of Responses	Percentage Reporting
Lethargy/Listlessness	347	88.3%
Decreased appetite	308	78.4%
Weight loss	282	71.8%
Fever	258	65.7%
Refusing to eat (anorexia)	188	47.8%
Enlarged abdomen	167	42.5%
Hiding/Avoidance behavior	147	37.4%
Fluid involvement	116	29.5%
Difficulty walking	103	26.3%
Neurological involvement	103	26.3%
Difficulty breathing or coughing	95	24.2%
Jaundice	53	13.5%

**Table 5 animals-11-02257-t005:** Brand of GS-441524 reported. Many respondents used more than one brand during treatment.

Brand of GS (Concentrations)	Count of Responses	Percent
Mutian (multiple dosages and formulas)	76	19.13
Multiple Brands *	71	18.11
Shire (15 mg/mL)	62	15.82
Capella (15 mg/mL)	38	9.69
Hero—White Cap (17 mg/mL)	38	9.69
Other *	36	9.18
Oscar (15 mg/mL)	32	8.16
Hero—Blue Cap (15 mg/mL)	20	5.1
Pine (15 mg/mL)	19	4.85
Total	392	100.00
Not Reported	1	--

* Other brands reported: Ajax, Andy’s, Aura, Beat, Blossom, Brava, Dawn, Kitty Care, Lucky, Miner, Nina, Ocean, Phoenix, Rainbow, Rainman, Ruby, Sak, Spark.

**Table 6 animals-11-02257-t006:** Adjunctive therapies reported for cats treated with GS-441524. Owners were able to list more than one source, so the total number of responses is higher than the total number of respondents.

Supportive Therapies	Count of Responses	Percentage Reporting
Vitamin B1 or B12	138	41.8%
Steroids	125	37.9%
Other vitamins and supplements	103	31.3%
Fluids (SQ or IV, not specified)	90	27.4%
Antibiotics	74	22.4%
Light therapy	39	11.9%
Anti-nausea medication	36	11.1%
Appetite stimulants	34	10.3%
Gabapentin	27	8.2%

**Table 7 animals-11-02257-t007:** Complications and adverse events reported for cats treated with GS-441524. Owners were able to list more than one source, so the total number of responses is higher than the total number of respondents.

Complication	Count of Responses	Percentage Reporting
Vocalization with injection	309	82.0%
Pain at injection site	287	76.1%
Cat struggled during administration	236	62.6%
Scars/scabs anywhere	203	51.7%
Sore/wound at injection site	157	42.2%
Bleeding at injection site	147	39.4%
Increased activity	133	35.5%
Swelling at injection site	96	25.7%
Increased appetite	54	14.3%

## Data Availability

The data presented in this study are available on request from the corresponding author.

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
