# Peer review of "Unlicensed GS-441524-Like Antiviral Therapy Can Be Effective for at-Home Treatment of Feline Infectious Peritonitis"

_animals, 2021, doi:10.3390/ani11082257_

Round 1

Reviewer 1 Report

I found this paper to be original, interesting and well written. However, the first thing that i believe needs to be mentioned in the abstract as well, not just in the discussion is the fact that the diagnosis of FIP is only a suspected one, or presumed, and by no means definitive (as some cats were not diagnosed by their vet (line 83, 84).

  1. Were the criteria by which cats were determined to be infected with FIP dependent on blood tests and clinical symptoms alone? lines 83, 84. You answer this in line 366, but this should be mentioned earlier. I would even say presumed FIP. Still it begs the question, by which criteria?
  2. Is there any information regarding the clinical stage of the presumed disease? 

3. How were cats that were not treated by the vet diagnosed? (line 138)

4. Line 160: who decided on the treatment protocol and how? Was it by the website administrators or the owners? (line 277)

5. Line 190: what was the time period in which the cats were reported to be alive or deceased?

6. Line 204 and 342: please clarify the meaning of “single relapse”

7. Please elaborate on what “complete remission” means. Complete resolution of some or all of the clinical symptoms? For how long? (line 293). Were cats that showed improvement but no complete remission also included?

I find that the reference to the relatively low effectiveness of the treatment of the CNS and eye form is very interesting.

the paragraphs in the discussion regarding the limitations of this study are accurate and well presented. 

Reviewer 2 Report

This manuscript describes a survey of cat owners who utilized a black-market, illegally-obtained product, GS-441524 to treat their pets with feline infectious peritonitis.  While the work of Pedersen undeniably shows the efficacy of GS-441524 for cats with FIP, the decision of Gilead to not market their product as an FDA approved drug precludes the ability of veterinarians to offer this solution to owners.  As such, any use of the drug is illegal and the use of social media groups to prescribe treatments likely also classifies as practicing of veterinary medicine without a license, a punishable offense.  Further, in many locations, the possession of needles is not without regulation, yet is also often required in order give this unlicensed product.  The use of a black market product, by nature, does not guarantee efficacy or potency and further encouraging such use can have serious consequences for owners and their pets who end up with an adulterated product.  Veterinarians should strive to practice evidence based medicine, but encouraging illegal activities is not in the best interest of those who have dedicated their lives to saving animals.  Further, the potential for GS-441524 to drive resistance has not been formally evaluated and the unregulated use of the product may also create additional consequences, especially when used by pet owners who lack the formal training to diagnose their animal.  Veterinarians are fully aware that this product exists and are aware of the potential efficacy of such drugs, but should not be encouraged to participate in illegal activities, putting their license and livelihood at stake.  Overall, this manuscript carries serious ethical concerns in regards to encouraging an illegal product and should be rejected.   

Additional major concerns of this manuscript include:

  1. The authors have no way to confirm that the cats reported on in this study were true FIP cases. As such, the manuscript lacks the ability to truly demonstrate an effect of these products. While likely that many of these cats were indeed FIP cases, it cannot and should not be assumed. 
  2. This study suffers from selection bias. Asking individuals in a social media group about a product they are encouraging the use of will inevitably lead to positive highlights of said product. 
  3. The use of this compound raises significant welfare concerns. These cats are in pain and not being appropriately managed by a veterinarian in addition to owners not being trained in proper administration of an injectable product.
  4. Line 357 – Regardless of power, this study cannot accurately say if any of the products are better or worse without a blinded observer, such as a veterinarian, to assess disease improvement. As such, such an assessment should be performed via a randomized controlled trial.   

Minor concerns in this article include:    

  1. The introduction could benefit by including additional references, including in regards to the use of Remdesivir with COVID-19 (line 44).
  2. Please correct the manufacturer’s name (line 45).
  3. Line 51 please define the first use of FCoV
  4. Lines 62-63 – Why Gilead stopped marketing the drug is irrelevant and painting Gilead as the bad guy doesn’t help move the conversation towards making the drug regulated and affordable.
  5. Lines 91-92 – It would be helpful to know if the authors decided on this criteria before or after they looked at their survey and how they formally considered a survey not usable. If 51% of the survey was completed, would it be eligible for inclusion?
  6. It would be helpful for the authors to provide a copy of their survey. For instance, in table 3, were these the only choices that respondents were given or was this an open ended response? Additionally, I am hesitant to believe that the authors are reporting all of the questions asked, as they note the survey took owners nearly thirty minutes and from what is reported, I would not believe it would take an owner this long.
  7. Line 111 – 12.6% is repeated in regards to intact males. The authors should follow a similar pattern for reporting sex across males and females
  8. Line 116 – How many cats were over 9 years of age and why was that cut off used? With reporting the age, it may be helpful to refer to the AAHA guidelines on age breakdown.
  9. The reported use of antibiotics in table 3 is not without concern for bacterial resistance and should be mentioned in the discussion. Information of where these antibiotics were obtained would be helpful.   
  10. Please correct the numbering for tables 6 and 7 to reflect what is being referred to in the manuscript.
  11. In regards to table 7, increased appetite or activity are arguably not adverse events.
  12. Line 214 – Please avoid the use of “To the author’s knowledge.” PubMed and other search engines should be used to conduct a thorough literature review. Please refer to: Di Girolamo N, Meursinge Reynders R. On "authors' knowledge" and contrast-enhanced ultrasonography in rabbits. Vet Radiol Ultrasound 2019;60:371. DOI: 10.1111/vru.12748.
  13. Line 325 – What is the relevance of a neural tube defect, a birth defect, in this report?

Reviewer 3 Report

An online survey of 393 people to determine the outcome of using unlicensed anti-coronavirus drugs used to treat feline infectious peritonitis (FIP) was presented.  This is a very interesting paper which will help many people and is certainly worthy of publication even though the authors appear to be inexperienced at writing scientific publications and still need to do quite a bit of work to bring it up to standard.

The authors found that the vast majority of people using anti-coronavirus drugs experienced recovery of their cats.  The writers found that 3.3% of cats died and 2.7% relapsed: these are useful figures.  Presenting the average cost was also useful for people proposing to embark upon this venture.

The most striking – indeed shocking – finding was that only 8.7% / 26.5% of cat lovers claimed to receive significant help from their veterinarians.  The figure 8.7% appears in the abstract but further reading of the paper doesn’t really justify this initial claim: it is more like 26.5%.  Nevertheless this shows a massive dereliction of duty by the veterinary profession who left 73.5% of cat guardians to fend for themselves faced with a traumatic diagnosis.  The authors of this paper have done very well to highlight this problem and I hope they may do a further study of the vets themselves to find out what made such a usually caring profession be so uncharacteristically negligent.  The authors speculate on the reasons in the discussion section of their paper, but it would be nice to have evidence, especially since most cats were in the USA – were vets afraid of being sued by the clients? Or was it mostly fear of the law, or the veterinary regulatory bodies?

Do you have sufficient data to construct Kaplan Meier curves for the various brands you document? If so, that would take your paper to a higher standard.

I have one major concern about the study: were you able to ascertain whether or not people were able to submit multiple replies, e.g. by verifying their IP addresses?  If the owners of the Facebook pages had a vested interest in selling their products, could they have interfered with your results? 

Your conclusion paragraph was outstanding!  Well done!

This is a paper which the public and veterinary profession will welcome, but it requires major revision.

MAJOR CRITICISMS

First of all, the term “black market” is too inflammatory and should be replaced throughout with the word unlicensed or something similar: you could say that they’re not registered with the FDA PROVIDED you confirm that with the FDA: conjecture is unacceptable in scientific publications. (Keep black market as a keyword though: it is in common use.) The lack of verification of your statements is a problem throughout your paper – see the comments about references and your rather cavalier classification of the online anti-virals.  

You have lumped together a disparate group of drugs and called them GS-441524 as if they were all the same: they are not.  Some are nucleoside analogues, others may be 3C-like protease inhibitors (see Pedersen 2018).  Some are injectables (e.g. Mutian II), some are pills (e.g. Mutian X), but you have not differentiated that.  You need to make clear throughout what people are dealing with – perhaps replace GS-441524 with “online anti-virals” or some other term.  You cannot call something GS unless it was made by Gilead Sciences, that’s what GS stands for, and if somebody else is calling a drug GS-441524 then you need to explain that it is wrongly named or that the name has become a catch-all for various anti-virals.  Find out if Hero etc are nucleoside analogues or something else.  Can you provide some sort of source for the drugs you cite?  For example, in their papers Addie et al (2020) gave the source for Mutian Xraphconn pills as Nantong Biotechnology, China, just as Pedersen et al, 2019 gave their sources of GS-441524 as Gilead Sciences.  Addie et al also specified that Mutian pills were adenosine nucleoside analogues: you need to be as specific as she was about what is in each of the makes. 

Line 12: remove the word illegal here and throughout.  First of all, none of you appears to have any legal qualifications, and even if you did, your qualifications would be pertinent only to your own country and 26% of your survey respondents came from a variety of non-USA countries, (including China where clearly the drugs are probably not illegal, since they seem to be mostly made there): do you claim to be experts in the laws of the whole world?   Unless you have legal expertise and have thoroughly investigated the laws pertinent to these drugs, it is an area you would do well to leave alone, although I believe that you are correct in stating that any GS-441524 based drug sold in the USA would be in contravention of Gilead’s patent, but I also have no legal education so cannot cite the relevant law and am also guilty of speculation.  I like your sentence which begins in Line 70 and wouldn’t want you to delete it, but tone it down to accurately reflect the situation: that veterinarians are uncertain of the legality of using these drugs because they are unlicensed with veterinary authorities (or the FDA or whatever) … therefore …  As I said above, you have done well to bring to attention the failure of our profession to adequately care for our patients.

Provide accurate and original references throughout: use Pubmed more.  Lines 47, 50  require references.   Line 51: reference 1 isn’t sufficient: please use source papers rather than reviews. In addition, your placement of the reference is confusing: are you saying that 90% of the cats are obtained from breeders, shelters, etc. or are you saying they’re seropositive?  A prevalence of 50% seropositivity in single cat households seems abnormally high:  please provide an original source reference.  Another reference you’ve wrongly attributed is that the figure for transiently infected and carrier cats came from Addie & Jarrett, 2001.

Please supply legends to explain all of your tables: e.g. percent of what?  It’s obvious to me, but you cannot assume it’s obvious to everybody.  Tables should be able to standalone – explain them in the legend, as if to somebody with absolutely no knowledge of the subject.

Lines 85-86 and 177: “most of” and “some” are not acceptable in a scientific paper – you must state the numbers accurately.  This criticism MUST be remedied throughout the paper prior to publication.

MINOR CRITICISMS

Another word you might want to reconsider is “owner:” can a sentient being really be owned?  It’s a word that is destined for the dustbins of history, just as the words “slave owner” are now deemed unacceptable, despite the rise in slavery due to recent wars in Libya and the middle east.  Keep the term if you wish – this is just a suggestion.

Line 25: which online group?  Be specific or tell the reader why you are not being specific.

Line 62: again you are speculating instead of doing the research: GS-441524 was previously tried on Ebola virus.  You might want to have a look into Gilead: it has a track record somewhat like Monsanto:  ethical standards that would embarrass a tapeworm.

Lines 81 & 93:  it is customary to provide the advertisement and survey you used in supplementary material in studies such as yours.  These should be provided so that your study can be replicated if anybody should wish to and so that reviewers can check if the survey questions are leading, or flawed.

Line 83: this is too vague – how were FIP diagnoses confirmed?  How many diagnoses were by veterinary surgeons and how many by the cat owners themselves?  How many responses did you exclude from your study on the basis that FIP was not confirmed, and of those, what was the outcome?

Line 90: you may have excluded useful information: you should include an analysis of the cats who had not had a full 84 days of treatment separately – had they been misdiagnosed?  Did they die? Recover?  A published case report described a cat who recovered after only 7 weeks of Mutian pills so that 12 weeks is not always necessary.

Lines 114-117: make sure you understand the difference between mean and median then check this sentence please: perhaps provide a bar chart of the cats’ ages?

Line 119 – 121: repetitive of “The remaining 32.8% were a wide 121 variety of pure breeds,”

Line 123 doesn’t agree with your statement in the abstract – in fact, 26.53% veterinary surgeons directly or indirectly helped the cat’s family to find a treatment.  You need to give numbers with percentages in brackets throughout your paper, not just percentages.

Table 2: percentage of the purebred cats or of all cats? – explain in a legend?  It’s customary to call non-purebred cats “domestic” – mixed can imply a cross between two purebreds.

You need to put totals in your tables, for example Table 3 only adds up to 392, not 393.

Line 139: can you give any indicators of which costs correlated with brands used?

Table 3: please give further details about the non-Facebook websites: for example, Prof. Pedersen’s papers were open access – how many people found his (or another) paper directly?  How many found treatment via the Pubmed website?

Table 4: give absolute numbers.  Fluid involvement???  Did you differentiate between ascites and pleural effusion?

Lines 148 -153 (and throughout): give numbers please.

Line 164: don’t you mean “increased” not decreased?  Pedersen et al reported that the cats gained weight therefore doses should have been increased as the cats got heavier.

Doses: it doesn’t seem reasonable to lump all the dosages together for disparate drugs as you have done: please add dosages to Table 5 or make a new table, and put the outcome and numbers, as below.

Table 5: include the outcome for each of these products please.  If you can, describe the contents of each of the makes you list, and differentiate between Mutian injections and Mutian pills.   What is the difference between Hero White Cap and Hero Blue Cap? 

Were none of the cats treated with Aura?

Table 5: there are two periods after the title.

Table 6: were none of the cats given antibiotics?  Please add numbers and an outcome column to this table.

Line 180: Tables 6 and 7 are mixed up in the text.

Table 7: how is increased appetite an adverse event in a disease in which anorexia is a clinical sign?

Line 190: at last some numbers!  But the wrong way round: put the numbers then the percentages in parenthesis.

Line 193: you need to give how many cats were in the effusive and non-effusive groups.

Lines 229-233: your reference number [4] is in the wrong place – you’re implying that the reference provides evidence for the cat owners being more affluent, whereas it’s evidence of the purebred prevalence in FIP cases.  You MUST become more accurate with your reference placing.

Line 248: was SDMA tracked?  Addie et al reported a rise in SDMA in their case report.

Lines 252-255: well done.  An excellent sentence.

Did you ask whether insurance companies paid for treatment?

Line 260: you make another very good point.

Line 265: you misrepresent Mutian: I saw a Facebook post where a Mutian employee speculated that Gilead had analysed their product and found it not to be GS-441524, because their lawyers had sent cease and desist letters to other makers, but not to Mutian.  GS-441524 is an injectable, Mutian is effective orally, and unlike GS injections, it totally clears the virus. 

Line 274: see Addie et al, case report (reference below).

Line 285: replace black market GS-441524 with “online antivirals.”

Line 285: put in reference.

You might want to read Choi et al, 2020 to see how Remdesivir fails to clear SARS-CoV2 infection.

Line 295: replace the word “rogue” with “caring.”  Bear in mind that Prof. Pedersen might be amongst their number!   

Line 302: what conflicts of interest do the group administrators have, do you know?

Line 366: you state there’s no way to confirm FIP diagnosis other than biopsy, but nowadays positive FCoV RT-PCR on effusion of tissue FNA is considered acceptable confirmation: see Tasker et al, 2018 and the Pedersen papers.

Line 373: this is a courageous suggestion, I hope the other reviewers and editor will allow it to stand though, but I would request you add the proviso that the cat not be suffering from kidney or advanced liver disease, since nucleoside analogues are renotoxic and hepatotoxic.  There is precedent for this approach in toxoplasmosis where often diagnosis is by assessing response to treatment.

Line 415: did you ask if they’d used an anti-viral in humans?

As I said above – well done re your last paragraph!  Though please modify the GS terminology as requested elsewhere.

References

Addie DD.  Curran S, Bellini F, Crowe B, Sheehan E,  Ukrainchuk L, Decaro N.  2020. Oral Mutian® X stopped faecal feline coronavirus shedding by naturally infected cats.   Res. Vet. Sci. 130:222-229.. doi: 10.1016/j.rvsc.2020.02.012.

Addie DD, Covell-Ritchie J, Jarrett O, Fosbery M. 2020. Rapid Resolution of Non-Effusive Feline Infectious Peritonitis Uveitis with an Oral Adenosine Nucleoside Analogue and Feline Interferon Omega. Viruses. 12; 1216. doi:10.3390/v12111216

Addie, DD, Jarrett, JO. 2001. Use of a reverse-transcriptase polymerase chain reaction for monitoring feline coronavirus shedding by healthy cats.  Vet. Rec. 148:649-653.  

Choi B, Choudhary MC, Regan J, Sparks JA, Padera RF, Qiu X, Solomon IH, Kuo HH, Boucau J, Bowman K, Adhikari UD, Winkler ML, Mueller AA, Hsu TY, Desjardins M, Baden LR, Chan BT, Walker BD, Lichterfeld M, Brigl M, Kwon DS, Kanjilal S, Richardson ET, Jonsson AH, Alter G, Barczak AK, Hanage WP, Yu XG, Gaiha GD, Seaman MS, Cernadas M, Li JZ. Persistence and Evolution of SARS-CoV-2 in an Immunocompromised Host. N Engl J Med. 2020 Dec 3;383(23):2291-2293. doi: 10.1056/NEJMc2031364.

Pedersen NC, Kim Y, Liu H, Galasiti Kankanamalage AC, Eckstrand C, Groutas WC, Bannasch M, Meadows JM, Chang KO. 2018. Efficacy of a 3C-like protease inhibitor in treating various forms of acquired feline infectious peritonitis. J. Feline Med. Surg.  20 (4): 378-392.

Tasker S.  2018.  Diagnosis of feline infectious peritonitis: Update on evidence supporting available tests.  J Feline Med Surg. 20(3):228-243.

Reviewer 4 Report

Thank you for an interesting article on use of GS-441524 by owners for the treatment of presumed FIP. I agree with the authors that while unconventional, it is important to publish this type of information so that veterinarians are aware of what is happening and to guide future clinical trials.

I am pleased that the authors identify some of the major limitations of their study with regard to owner reported diagnosis of FIP, and the fact that some of the cats may not have had FIP. The authors also briefly mention the fact that the products used may have varied in their GS-441524 content. I would encourage the authors to take this one step further, because it is possible that some of these products did not contain any GS-441524 at all (this has certainly been reported with other black marked products).

Given these limitations  I think that the authors need to be more conservative in their conclusions. For example using terms like “owner reported success/cure”, or “apparent success/effectiveness/cure/remission” (in the title and conclusions of the paper), rather than “can effectively cure”.

My additional comments are listed below.

Abstract:

  • Please add a line or two to your simple summary and abstract to clarify some of the limitations of your study including relying on owner reported diagnosis of FIP, owner reported clinical signs, owner reported cure and relapse, and inability to determine compliance with treatment protocols and whether or not the products contained the active ingredient at all or in the quantity suspected
  • Please also include a line on the most commonly reported adverse events in your simple summary and abstract

Introduction:

  • Line 61 – please provide a reference here
  • Line 66 – please provide a reference here
  • Can you please clarify why this drug is illegal – I presume that this is obvious (ie. it is not approved by the FDA and/or being used without a prescription), but I do believe it needs to be clarified

Materials and Methods:

  • Line 81-83 – can you please reference the social media page that was used to distribute the survey
  • Can you please include the survey as an Appendix to the article so that the reader can clearly see the questions that were asked
  • Was there any way to ensure that two people from the same household didn’t answer the survey about the same cat (eg. removing duplicate IP addresses)
  • Line 85 – here you mention “the full 84 days of treatment” for the first time – can you please include some information in the introduction about published treatment protocol(s) with this drug including the route of administration, dose, dosing frequency and duration of treatment (at least that was used in the published study be Pederson et al.) – some of this is buried in the results (eg. lines 166 to 168) and discussion, but I think it just needs to be clarified up front
  • When you describe published treatment protocols please also clarify the observation period
  • Regarding your statistical methods, did you assess whether the data was normally distributed before you used parametric tests?

Results:

  • Line 116 – was there a bimodal distribution of ages as is commonly described for FIP
  • Can you please adjust the table titles so that they stand alone. Additionally, some tables are in alphabetical order while others are from most common to least common – I would recommend standardising them – firstly from most common to least common (and then in alphabetical order eg. for the cat breeds represented only once)
  • Also for the tables can you please clarify the number of respondents that answered the particular question, so that we know what numbers the percentages translate to
  • Line 138-141 – presumably this is in USD (as stated in the discussion) but it needs to be clarified here (or in the methods)
  • Did you ask questions about any other medical conditions in the cats? Presumably with such a large sample size some of the cats must have had other medical conditions – please clarify
  • Consider using a flow chat to display outcome since as written this is a little confusing. Please also include the actual number of cats (n) as well as %

Discussion:

  • Please be more explicit of the limitation that the products thought to contain GS-441524 may not have contained this drug at all
  • Please also be more explicit that owner reported clinical signs eg. neurologic and ocular signs may not have truly been neurologic and ocular manifestations of FIP
  • Please clarify that owner reported “cure/remission” or “relapse” may not have been true cure/remission or true relapse
  • Please clarify whether or not you were able to differentiate what type of steroids – presumably you mean glucocorticoids, but it may be possible that owners used anabolic steroids if their cats weren’t eating
  • Line 402 – the Chinese study here receives only a cursory mention. Can you please elaborate on the findings of the design and findings of this study
  • As requested in the beginning of this review, please reword so as not to overstate your conclusion based on the aforementioned limitations

Round 2

Reviewer 2 Report

None

This manuscript is a resubmission of an earlier submission. The following is a list of the peer review reports and author responses from that submission.